# Crystallization, Structures, and Properties of Different Polyolefins with Similar Grafting Degree of Maleic Anhydride

**DOI:** 10.3390/polym12030675

**Published:** 2020-03-18

**Authors:** Ying Wang, Ying Shi, Wenjun Shao, Yi Ren, Wanyu Dong, Farao Zhang, Li-Zhi Liu

**Affiliations:** 1Advanced Manufacturing Institute of Polymer Industry, Shenyang University of Chemical Technology, Shenyang 110142, China; wy584436393@163.com (Y.W.); shiying@syuct.edu.cn (Y.S.); shaowenjun0813@163.com (W.S.); renyi86@163.com (Y.R.); dwy839085372@163.com (W.D.); 2Ningbo MaterChem Technology Co, Ltd., Ningbo 315830, China; zfarao@materchem.com

**Keywords:** MAH grafted polyolefins, crystallization behavior, crystal structure, mechanical properties

## Abstract

Maleic anhydride (MAH) grafting to different polyolefins with similar grafting degree can have different effects on crystallization, crystal structure, and mechanical and thermal properties. The grafting leads to a smaller crystal size, less ordered lamellar structure, and a shorter long period for HDPE, LLDPE, and PP. The grafting makes PP lamellar packing less ordered the most and almost no effect to LLDPE. The grafting does not have that much impact on the crystallization ability of the HDPE, LLDPE, and HDPE/PP blend, but appreciably reduces the crystalline ability of PP-*g*-MAH, due to a dramatical drop in its molecular weight during the grafting process. As a result, the grafting makes PP a very brittle material with a lowered average melting point than the corresponding neat PP, but the grafting has almost no effect on elongation at break for LLDPE and some effect on HDPE (decreased by one-third). However, the PP degradation due to MAH grafting can be avoided in the presence of PE component, i.e., making the grafting of PP and PE at the same time with HDPE/PP blend. The grafted HDPE/PP blend shows a significantly improved compatibility, which leads to overall appreciably better mechanical properties than the neat HDPE/PP blend.

## 1. Introduction

Maleic anhydride (MAH) grafted polypropylene (PP) and polyethylene (PE) are semi-crystalline polymers with a wide range of applications in industry [1,2], such as chemical coupling agents [3], impact modifiers [4], and reactive compatibilizer precursors [5] for blends. Maleic anhydride grafted polypropylene (PP-*g*-MAH) and maleic anhydride grafted polyethylene (PE-*g*-MAH) have received lots of attention due to their extensive use in improving the interfacial interaction between components in polymer blends and polymer composites in order to maximize physical properties [6]. Some studies related to the blend of MAH grafted polyolefins with Nylon 6 (PA6), clay (montmorillonite), etc. have also been conducted [7,8,9,10,11].

In the early days, many studies were carried out on the crystallization behavior of polyolefin grafted with MAH. The work done by Seo et al. [12] shows that PP-*g*-MAH can play a role as a nucleating agent in iPP/PP-*g*-MAH blend. Harper et al. also studied PP/PP-*g*-MAH blend, as well as the corresponding neat PP [13]. Their work on the blend suggested that the MAH grafting led to a lowered melting point of PP component, likely due to the co-crystallization of PP and PP-*g*-MAH. Compared with the neat PP homopolymer, the dynamic storage modulus of the blend increased slightly with addition of small amount of PP-*g*-MAH, while mechanical dampening decreased with addition of PP-*g*-MAH. 

It is well known that the degree of MAH grafting can have significant impact on the crystallization and structure of polyolefins. The melting temperature of HDPE-*g*-MAH decreases as increase in grafting content from 0.2% to 0.8%, [14] but increases with further increase in grafting degree. The crystallization, melting behavior and structures of PP with different maleic anhydride contents (AC = 0.5 mass%, 1 mass%, 3 mass%) was investigated by Menyhárd et al. [15] Liu et al. prepared and studied maleic anhydride/styrene-grafted PPR (polypropylene random copolymer) with grafting degrees of 1.38%, 2.25%, and 2.42%, respectively, and it is found that the crystallization rate of this MPP is higher than the neat PP and increases with the increase in grafting degree [16].

Although there are quite a number of studies on MAH grafted polyolefins—including PP [15,16], high-density polyethylene (HDPE) [14], and linear low-density polyethylene (LLDPE) [17]—the different effects of similar MAH grafting degree on crystallization, structures, and properties of different polyolefins have not been explored. In the present work, a series of MAH grafted polyolefins are studied, including maleic anhydride grafted high-density polyethylene (HDPE-*g*-MAH), maleic anhydride grafted low-density polyethylene (LLDPE-*g*-MAH), PP-*g*-MAH and maleic anhydride grafted polypropylene/maleic anhydride grafted high-density polyethylene (PP-*g*-MAH/HDPE-*g*-MAH) blends (1:1). The crystallization kinetics, thermal behavior, crystal structure, lamellar packing, and crystal size of these grafted polyolefins, together with the corresponding non-grafted polymers, are discussed, which provides a more in-depth understanding about the structure–property relationship of the MAH grafted polyolefins. On the other hand, though double melting points of PP-*g*-MAH were reported in several studies in literature [15], it is not well understood what leads to the two distinct melting peaks for the grafted PP. In literature, it is interpreted as the perfection of the crystal structure during the melting process [15]. With the current study, a very different conclusion is made about the double melting points based on our studies with multiple techniques, including degradation characterization [18].

## 2. Materials and Methods

### 2.1. Materials and Sample Preparation

Commercial PP, FH118, LLDPE, MC226, HDPE, GPM128 PP/HDPE (1:1), GPM135 were obtained from Ningbo Mater Chem Technology Co, Ltd., Ningbo, China. FH118, MC226, GPM128, GPM135 were prepared by PP, LLDPE, HDPE, and PP/HDPE (1:1) melt grafted MAH, denoted as PP-*g*-MAH, LLDPE-*g*-MAH, HDPE-*g*-MAH, PP-*g*-MAH/HDPE-*g*-MAH. Among them, LLDPE is ethylene and butyl copolymer. Melt index and grafting degree for each material are listed in Table 1.

### 2.2. Thermal Analysis

Melting and crystallization behavior of the blends were characterized by a TA Q100 Differential scanning calorimetry (DSC) in nitrogen atmosphere. The sample was heated to 200 °C at a heating rate of 10 °C/min, holding for 5 min, and then cooled at the same rate to −20 °C. The samples were then heated back to 260 °C at a rate of 10 °C/min for the study of melting behavior. The normalized crystallinity (X_c_) of PP or PE component was determined by X_c_ = (ΔHm/ΔH^0^m)/φ, where φ is the weight fraction of PP or PE. The theoretical heat of fusion ΔH^0^m for 100% crystallized polypropylene [19] and polyethylene [20] are 207 J/g and 293 J/g, respectively.

### 2.3. Synchrotron Wide-Angle X-ray Diffraction (WAXD) Measurements

The pre-weighed pellets were evenly placed in a mold. This polymer was melt-pressed at 200 °C, 5 MPa for 5 min to make a sheet of 1 mm thickness and subsequently cooled down to room temperature without releasing the pressure. All specimens were cut into rectangular strips of 0.1 × 1 × 1 cm^3^ dimensions.

Synchrotron wide-angle X-ray diffraction experiment was performed using synchrotron radiation with *λ* = 0.154 nm at Beamline 1W2A of Beijing Synchrotron Radiation Facility (Beijing, China). Mar165-CCD was set at 96.25 mm sample-detector distance in the direction of the beam for WAXD data collections. The exposure time was 11 sec. The collected data were corrected for air background before any analysis.

### 2.4. Small Angle X-ray Scattering (SAXS)

The samples are prepared in the same way for WAXD, as a sheet of 1mm thickness.

Small angle X-ray scattering (SAXS) measurements were performed at Xenocs X-ray small angle scatterometer (Xeuss 2.0, Grenoble, France). The scattering intensity was detected by Pilatus3R 300 K detector with 487 × 619 pixels and 172 μm pixel size. The data were collected with Genix3D Cu operated at 50 kV and 0.6 mA (30 W). The wavelength (*λ*) used was 0.154 nm. The detector area is 83.9 × 106.5 mm^2^ and the spot size is 0.8 × 0.8 mm^2^. The sample-detector distance is 2500 mm. The exposure time was 300 sec. The position of the peak *q_max_* is related to the long period *L* by the Bragg’s law: *L* = 2*π/q_max_*, where *q* is the scattering vector, is defined as *q* = 4*π* (sin *θ*) /*λ*, where *λ* is the X-ray wavelength, *θ* is half of the scattering angle (2*θ*) [21]. The collected data were corrected for air background before any analysis.

### 2.5. Tensile Testing

The same method was used to prepare a sheet having thickness of 2 mm, using a cutting machine dumbbell-shaped sample bars with dimensions of 115 mm of length, 25 mm of neck length, and 6 mm of neck width.

A universal testing machine (Instron 3365, Boston, MA, USA) was used to test the tensile properties. Each sample was clamped at both ends and then stretched at a crosshead speed of 30 mm/min. The initial length l_0_ is 25 mm. At least five specimens were tested for each sample to obtain a reliable average and standard deviations for all the mechanical properties. All the mechanical properties were performed at room temperature. From the stress–strain curves, the following properties (based on averages of four samples) were calculated: Young’s modulus (from the initial slope of the stress–strain curve), yield strength, and yield elongation (from the first maxima of the curve), elongation at break where the sample fails.

## 3. Results and Discussion

MAH grafting on polyolefins have a significant impact on crystalline structure and properties (mechanical and thermal). The effects can be quite different for different polyolefins-*g*-MAH. In the present paper, the effects of MAH grafting on crystallization kinetics and melting behavior are studied with DSC, and the corresponding effects on crystalline structures and crystal packing structure are studied with WAXD and SAXS. The effects of MAH grafting on mechanical properties of different polyolefins with similar MAH degree are studied with an Instron tensile stage.

### 3.1. MAH Grafting Effect on Crystallization Dynamics and Crystal Size for Different Polyolefins

Non-isothermal crystallization of these materials was studied with DSC by a heating, cooling, and reheating process. Crystallization temperature (*T*_c_), melting point (*T*_m_), the enthalpy of melting (*H*_m_), and crystallinity are determined from the DSC curves. In terms of the effect of MAH grafting on the crystalline ability, very significant decrease is observed for PP-*g*-MAH. The cooling and second heating traces of the MAH grafted PP and pure PP are shown in Figure 1a. As it is seen from this figure that the crystallization temperature of PP-*g*-MAH decreases very significantly (about 5.8 °C) compared with the neat PP sample. Generally, PP with lower molecular weight shows a lower crystallization temperature [22]. Thus, the crystallization ability was significantly decreased due to a significant increase of the melt index of PP-*g*-MAH (Table 1). A significant difference is also noticed from the melting process as shown in Figure 1a for PP and PP-*g*-MAH. Only one melting peak is observed for the neat PP at 165.1 °C while two distinct melting peaks are observed for the grafted PP. One of the melting peaks for the grafted PP appears at the same temperature (165.3 °C) as the neat PP, suggesting that these crystals are formed with the PP molecules which have similar molecular weight as the neat PP and do not have many MAH branches. Another melting peak for PP-*g*-MAH appears at a significantly lower temperature (157.8 °C), which can be attributed to PP crystals composed of the molecules with lower molecular weights or may have more MAH branches. The crystallinity of the PP-*g*-MAH is larger than the neat PP; this is likely caused by a lowered average molecular weight due to the grafting. In fact, it is reported in literature [23] that as decrease in molecular weight of PP, a higher crystallinity degree is observed.

It is seen from Figure 1b that the LLDPE-*g*-MAH also shows a lower *T*_c_ than LLDPE but the difference is much smaller (0.79 °C) than the difference observed for the grafted PP and neat PP. The heating thermogram shows that the melting point of LLDPE-*g*-MAH is only slightly smaller than LLDPE, suggesting that the MAH grafts has very limited effect on the perfection and size of the crystals formed in the grafted LLDPE. In fact, the calculated crystallinity for LLDPE-*g*-MAH is about the same as the LLDPE (40.2%). The crystalline peak at 60 °C during cooling is supposed to be contributed by the small fraction of the LLDPE with high branching degree so that it cannot crystallize together with the major component around 105 °C. The crystalline peak around 60 °C for the LLDPE and the grafted LLDPE almost overlaps together, suggesting that the MAH are likely not grafted in the portion of LLDPE chain which already heavily grafted with short chain branching due to the copolymerization of ethylene and butene.

The thermograms for HDPE and HDPE-*g*-MAH obtained from DSC cooling and the subsequent heating are shown in Figure 1c. It is seen from this figure that the crystallization temperature during cooling process is almost same for HDPE and its corresponding HDPE-*g*-MAH. The effect of MAH grafting on the melting behavior is also very limited. Both the cooling and the heating curves indicate that the crystallinity of HDPE-*g*-MAH is slightly lower than the corresponding HDPE (62.7% vs. 64.9%). HDPE generally has some degree of short chain branching, which determines the density of the HDPE, though the branching degree can be significantly lower than LLDPE. As shown in Figure 1c, a small crystallization peak around 80 °C is observed during the cooling process of HDPE. This peak can be contributed to a small fraction of HDPE with more short chain branching, which cannot co-crystallize together with the major component at the temperature 117 °C. It is interestingly noted that such small crystallization peak for HDPE-*g*-MAH is at lower temperature, as shown in Figure 1c, suggesting that MAH can be grafted to the HDPE molecules with relatively more short chain branching.

The MAH grafting effect on PP, HDPE, and LLDPE is also reflected from crystal size obtained from X-ray diffraction. The crystalline structure and micro crystal size of the MAH grafted polyolefins and the corresponding non-grafted polymers was studied with X-ray diffraction, as shown in Figure 2. It is seen from this figure that diffraction peaks at 14.2°, 16.9°, 18.6°, and 21.8° are observed for the PP related samples, corresponding to the (110), (040), (130), and (131) crystal planes of α phase, respectively. That is, the PP-*g*-MAH still have alpha crystal form [24], though double melting peaks were observed from the DSC study above (Figure 1a). The diffraction patterns of LLDPE and HDPE are shown in Figure 2b and Figure 2c with (110) and (200) major indexes. It is noticed that MAH grafting has almost no effect on the diffraction patterns the materials (Figure 2), but the crystal size is different after the grafting, which can be evaluated based on Scherrer equation [25].
*L*(hkl) = K*λ*/(*β*cos*θ*),(1)
where *L*(hkl) is the crystallite size perpendicular to the crystal plane (hkl) direction, K is the shape factor (0.89), *λ* is the wavelength of the X-ray used, *β* is the full width at half maximum intensity (*FWHM*) of a diffraction peak in radians, and *θ* is the diffraction angle. As shown in Table 2, a smaller averaged crystal size a MAH grafted polyolefin than its corresponding non-grafted polymer is observed, indicating that the MAH grafts lead to the formation of smaller PP and PE micro crystals.

The discussion above shows that MAH has different effects on PP, LLDPE and HDPE. The T_c_ reduction degree reduces in the following order: PP, LLDPE, and HDPE. It may be depended on the microstructure of different polymer chains. The crystallization ability of PP-*g*-MAH during cooling is significantly lower than that of PP; its averaged crystal size becomes smaller than the neat PP. The average melting point of the grafted PP is lower than the neat PP, and its crystallinity is higher due to the decrease in molecular weight. MAH grafting also leads to a smaller average crystal size for LLDPE and HDPE, but has less impact on crystalline ability, presumably due to PE having a stronger crystalline ability.

### 3.2. MAH Grafting Effect on Crystal Packing for Different Polyolefins

Figure 3 shows the SAXS profiles of three different polyolefins grafted with MAH, together with the Lorentz-corrected profiles (inset figures). The linear SAXS profiles for PP and PP-*g*-MAH at room temperature are shown in Figure 3a, where a scattering peak is observed around *q* = 0.4 nm^−1^, corresponding to the long period for PP lamellar stacks. The scattering peak of PP is much better defined and has significantly higher scattering intensity than that of PP-*g*-MAH. The less ordered scattering from PP-*g*-MAH presumably due to the double melting peaks seen from DSC analysis (Figure 1a), which can be attributed to significantly reduced molecular weight for part of the PP-*g*-MAH molecules. The averaged inter lamellar distance (long period) of these two PP materials can be obtained from the Lorentz-corrected SAXS scattering profiles, as shown by the inset figure in Figure 1a. It can be seen from this inset figure that the scattering peak for the grafted PP locates at a larger *q* value than the PP, meaning a smaller average long period for the grafted PP. The evaluated long period for the grafted PP is 12.7 nm, significantly smaller than the non-grafted PP (15.1 nm), as shown in Table 3. The lower scattering intensity is also noticed for the grafted PP polymer (Figure 3a), which is believed to be caused by lower density difference between neighboring PP crystal lamella and amorphous lamellae due to increased density in amorphous region with addition of MAH grafts (density ~1.48 kg/m^3^).

Figure 3b shows a well-defined SAXS scattering peak for both HDPE-*g*-MAH and the corresponding HDPE, indicating that MAH grafting does not have that much effect on the lamellar packing, likely due to the strong crystalline ability of HDPE. However, a significantly lower scattering intensity of the grafted HDPE than the HDPE is also observed from Figure 3b, which is supposed to be caused by lower density difference between HDPE crystal lamellae and the amorphous HDPE lamellae due to increased density of the amorphous region with addition of MAH grafts (density ~1.48 kg/m^3^). The good lamellar packing for the two HDPE samples is also reflected from the Lorentz-corrected SAXS profiles, where the second order scattering is also observed for both polymers. The better-defined peaks for neat HDPE than the grafted HDPE (inset figures of Figure 3b) indicates the lamellar packing in the grafted HDPE is less organized than the neat HDPE. As also seen from inset figures of Figure 3b, a scattering peak shifting to a larger *q* value is also observed for the grafted HDPE. In fact, as listed in Table 3, the evaluated long period for the grafted HDPE is 24.4 nm, slightly smaller than the non-grafted HDPE (25.8 nm).

Less effect of MAH grafting on lamellar packing is observed for LLDPE, as shown in Figure 3c. The slightly broader scattering peak seen from the Lorentz-corrected profile for the grafted LLDPE suggests a little less well-defined packing structure in the grafted LLDPE then the neat LLDPE. The long period for the grafted LLDPE is 16.8 nm, also only slightly smaller than the neat LLDPE (17.7 nm). However, the scattering intensity of the grafted LLDPE is still reduced a lot due to a smaller density difference of crystalline and amorphous phases caused by MAH grafts.

That is, the SAXS study on the lamellar packing structure shows a less well-defined lamellar packing structure with a shorter long period is observed for the grafted PP, HDPE, and LLDPE than the corresponding neat polymers. MAH grafting makes PP lamellar packing less ordered the most, and then HDPE, and the least change in lamellar packing is observed for LLDPE.

### 3.3. Very Different Impact of MAH Grafting to Mechanical Property of Different Polyolefins

The stress–strain curves of LLDPE-*g*-MAH, HDPE-*g*-MAH, and PP-*g*-MAH and their corresponding neat materials are shown in Figure 4. It is seen from this figure that MAH grafting has little effect on the mechanical properties of LLDPE, as shown in Figure 4a and the summarized mechanical data in Table 4, which is consistent with the fact that MAH grafting has little impact to the lamellar packing structure of LLDPE as discussion in previous section. However, MAH grafting significantly decreases the elongation of HDPE. Specifically, the elongation at break is reduced from 2839% to 1856% for HDPE-*g*-MAH, as shown in Figure 4a. The tensile strength for the grafted HDPE also shows an appreciable drop (25.3 vs. 27.6, see Table 4), but the modulus of the grafted HDPE shows some gain (188 vs. 178, Table 4). As discussed above in SAXS section, MAH grafting makes the lamellar stacks of HDPE less defined with shorter long period, which is partially attributed to the reduced elongation. More significant impact of MAH grafting to mechanical property is observed for PP, as shown in Figure 4b. The elongation at break dropped dramatically from 1061% to 20.2%, together with some drop in tensile strength (from 25.6 to 16.6 MPa, Table 4). The poor mechanical property of the PP-*g*-MAH can be interpreted as follows. MAH grafting leads to a significant decrease in molecular weight, at lease, a portion of the PP, which in turn leads to some smaller crystals by exhibiting double melting peaks, 157.8 and 165.3 °C as shown in Figure 1a. From structure point of view, the evaluated long period for the grafted PP is 12.7 nm, a significant drop compared with the non-grafted PP (15.1 nm), meaning significant thinner crystal lamellae are formed in the grafted PP.

In summary, the effect of MAH rafting on mechanical property of different polyolefins is different. MAH grafting has a much greater negative effect on mechanical property of PP than PE. The grafting has a limited effect on the mechanical properties of HDPE, but almost no effect on mechanical properties of LLDPE.

### 3.4. MAH Grafting to PP/HDPE Bend (50/50) and the Grafting Effect on Compatibility and Properties of the Blend

It is well known that the PP and PE are not compatible, so in case of the application of PE/PP blending, compatibilizer may be needed. MAH grafted PP and PE could improve the compatibility of the two components. In the present work, the PE/PP blends are made first and then are grafted with MAH together. As discussed earlier, in the case of MAH grafting to PP, a significant decrease in molecular weight is observed, leading to the remarkable increase in melt index of the grafted PP to 64.3 g/10 min from 1.8 g/10 min for the neat PP, as listed in Table 1. However, it is noticed that the melt index of the grafted PE/PP blend (50/50) is very close to that of neat PE/PP blend (13.8 vs. 12.3 g/10 min), as shown in Table 1. This indicates that the significant decrease in molecular weight of PP due to MAH grafting can be avoided in the presence of PE component. This further suggests that the very significant drop in mechanical property of PP component can be avoided when the grafting is done together with PE component, i.e., conducting the grafting to PP/PE blend, instead of each component individually. On the other hand, the thermal property of the grafted PP component is also supposed to remains like neat PP. In fact, the study of the mechanical and thermal properties of the grafted blends and the neat blends confirms the above statement.

As shown in Figure 5 for the thermal study of the MAH grafted PP/HDPE blend and the corresponding PE/PP blend. One crystallization peak is observed at a cooling rate of 10 °C/min for PP-*g*-MAH/HDPE-*g*-MAH blend, and it locates at slightly lower temperature (Figure 5a). That is, at this cooling rate, the crystallization of PP and PE components cannot be differentiated. A DSC study on these two blends with a slower cooling rate at 2 °C/min was also conducted in the present work, as shown in Figure 5b. It can be seen from this figure that two distinct crystallization peaks are observed during the slower cooling process for the grafted blend and the un-grafted blend, with the crystallization at higher temperature corresponding to PP crystallization and crystallization at a lower temperature corresponding to PE crystallization. It is shown again that the MAH grafted blend crystallizes at a slightly lower temperature due to the grafting effect. Figure 5b also shows that narrowed double crystallization peaks are observed for the grafted blend, as the nucleation and crystallization start at appreciably lower temperature for the grafted blend.

Melting traces in Figure 5 for both cooling rates show two separated melting peaks with the higher melting peak corresponding to PP crystals and the lower melting peak corresponding to PE crystals. Specifically, the DSC curve of the PP/HDPE blend shows two melting points with peak temperatures around 134 °C and 165 °C, corresponding to the melting peak of PE and PP crystals, respectively. The DSC melting curve of PP-*g*-MAH/HDPE-*g*-MAH blend also has two melting peaks; the melting temperature for PE and PP crystals is 129 and 162 °C, respectively. Slightly lower melting temperatures for both PP and PE crystals for the grafted blend than the un-grafted blend are observed due to MAH grafting. The PP-*g*-MAH/HDPE-*g*-MAH blend also has a lower melting enthalpy of approximately 127 J/g, compared with PP/HDPE blend with melting enthalpy of 143 J/g, indicating a lower crystallinity for the grafted blend. As discussed before, double melting peaks are observed for PP-*g*-MAH with the lower T_m_ corresponding to the crystals formed by the PP molecules with a significant lower molecular weight caused by the MAH grafting. What needs to be addressed is that the thermal property of PP component in the grafted blend remains about the same as the neat PP, as the very significant drop in molecular weight of PP due to MAH grafting does not occur in the grafted blend.

SAXS study was also conducted for the two blends. As shown in Figure 6, since the two components in PP/HDPE blend are semi-crystalline polymers at room temperature, the SAXS scattering peak is contributed by both PP and HDPE lamellar stacks. It can be seen from the plots with linear intensity in Figure 6 that a much worse scattering peak is observed for the grafted blend, suggesting the lamellar packing in the grafted blend is not as good as in the corresponding un-grafted blend. In fact, for the Lorentz corrected profile, the second order of the scattering peak is observed for PP/HDPE blend, but absent for the grafted blend, also indicating a less ordered structure in the grafted blend. The less ordered lamellar packing in the grafted blend can be partially attributed to the improved compatibility of PP and PE components. The averaged interlamellar distance (long period) for the PP and PE crystals can also be obtained from the Lorentz-corrected SAXS profiles. The evaluated long period for the PP/HDPE blend and the grafted blend is basically about the same (23.8 vs. 23.4 nm).

Mechanical properties are very important for the grafted blend. The objective for making the grafted blend is to improve the compatibility of PP and PE for a better mechanical performance, while minimize the molecular weight drop in PP, which can negatively impact the mechanical properties. It can be seen from Figure 7 that the elongation for the grafted blend is about 2 times larger than the corresponding PP/HDPE blend (12% vs. 25%) with appreciable increased tensile strength at break. In fact, the tensile strength also shows a significant increase from 20.5 MPa for un-grafted blend to 27.5 MPa for the MAH grafted blend. The elastic modulus increases from 184.0 MPa for the un-grafted blend to 198.0 MPa for the grafted blend, due to the improvement of compatibility, as summarized in Table 5. That is, MAH grafting to PP/HDPE blend, not individually grafting, can improve the compatibility of PP and HDPE while not compromise with PP molecular weight, in order to improve the mechanical property of the blend.

## 4. Conclusions

In the present work, the effects of MAH grafting on LLDPE, HDPE, PP, and PP/HDPE blend with similar grafting degree were studied. From a structural point of view, the grafting creates a less ordered lamellar structure, a shorter long period for lamellar stacks, and a smaller crystal determined by X-ray diffraction than the corresponding neat polymers. MAH grafting makes PP lamellar packing less ordered the most and has almost no effect on LLDPE.

The MAH grafting does not have that much impact on the crystallization ability of the HDPE, LLDPE, and HDPE/PP blend, but appreciably reduces the crystalline ability of PP-*g*-MAH, due to dramatic drop in its molecular weight during the grafting process, as confirmed by the remarkable increase in melt index of PP-*g*-MAH.

The effect of MAH rafting on mechanical property of different polyolefins is different. MAH grafting has a much greater negative effect on mechanical property of PP than PE. The grafting has limited effect on mechanical property of HDPE, but almost no effect on mechanical property of LLDPE. It is the significant drop in molecular weight of PP-*g*-MAH during the grafting process that the PP-*g*-MAH becomes a very brittle material with a lower average melting point than the corresponding neat PP.

However, the significant drop in molecular weight of PP due to MAH grafting can be avoided in the presence of PE component. MAH grafting to PP/HDPE blend, not individually grafting to each component, can avoid the degradation of PP molecules. Therefore, the grafted PE/PP blend can be made as a good compatibilizer for PE/PP blends without compromising the mechanical and thermal properties of the compatibilizer.

The MAH grafted PP/HDPE blend shows greatly improved compatibility between HDPE and PP components. As a result, the elongation, tensile strength, and elastic modulus of the grafted PP/HDPE blend is significantly better than the corresponding un-grafted blend.

## Figures and Tables

**Figure 1 polymers-12-00675-f001:**
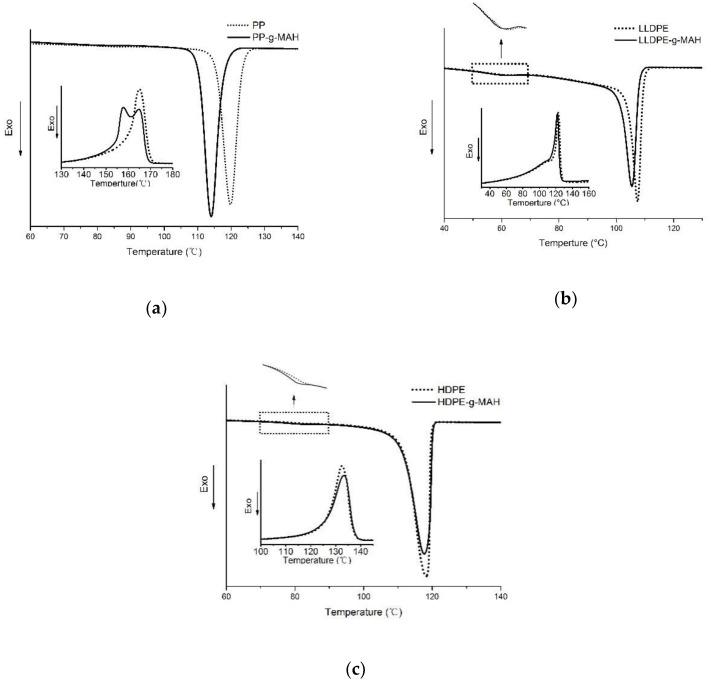
DSC scans of PP and PP-*g*-MAH (**a**), LLDPE and LLDPE-*g*-MAH (**b**), HDPE and HDPE-*g*-MAH (**c**).

**Figure 2 polymers-12-00675-f002:**
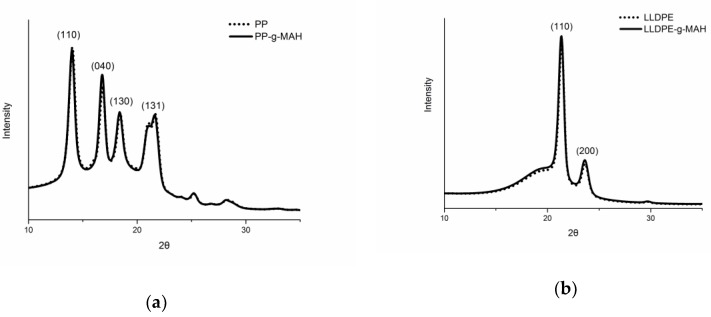
WAXS diffraction curves of PP and MAH-*g*-PP (**a**), LLDPE and LLDPE-*g*-MAH (**b**), HDPE and HDPE-*g*-MAH (**c**).

**Figure 3 polymers-12-00675-f003:**
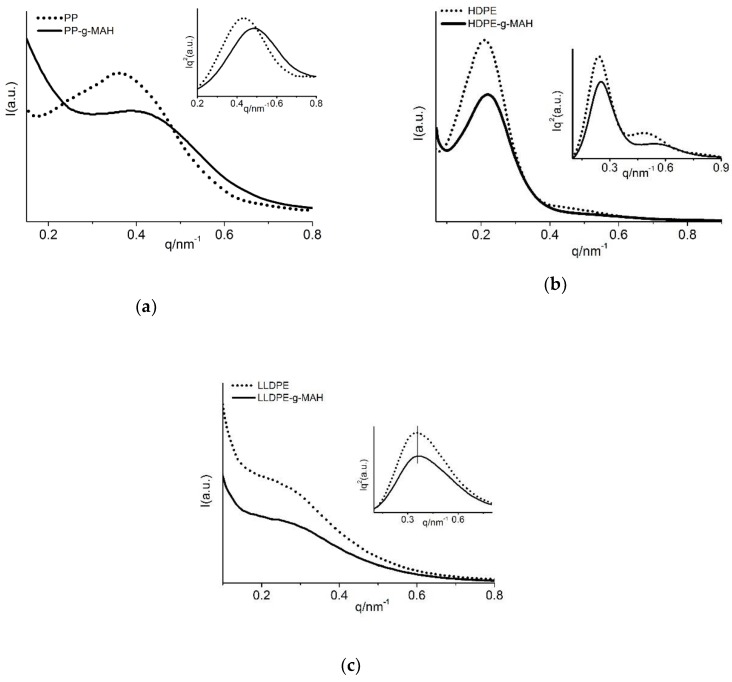
Linear and Lorentz-corrected SAXS profiles of different polymers. (**a**) PP and PP-*g*-MAH, (**b**) HDPE and HDPE-*g*-MAH, (**c**) LLDPE and LLDPE-*g*-MAH.

**Figure 4 polymers-12-00675-f004:**
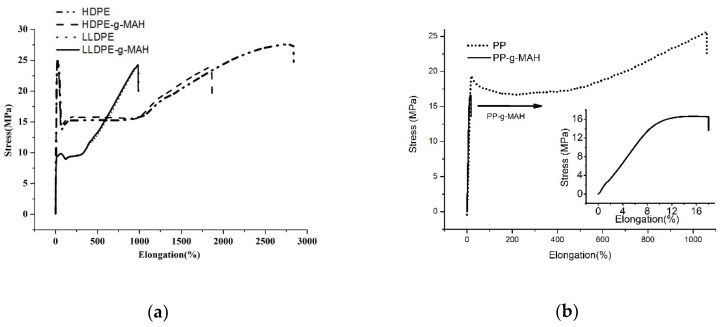
Stress and strain curves of (**a**) LLDPE, LLDPE-*g*-MAH, HDPE, HDPE-*g*-MAH; (**b**) PP and PP-*g*-MAH.

**Figure 5 polymers-12-00675-f005:**
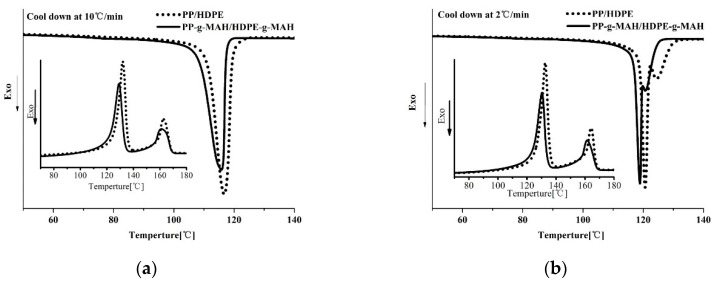
DSC thermograms of PP/HDPE blend and PP-*g*-MAH/HDPE-*g*-MAH blend at different cooling and subsequent heating rates: (**a**) 10 °C /min; (**b**) 2°C /min.

**Figure 6 polymers-12-00675-f006:**
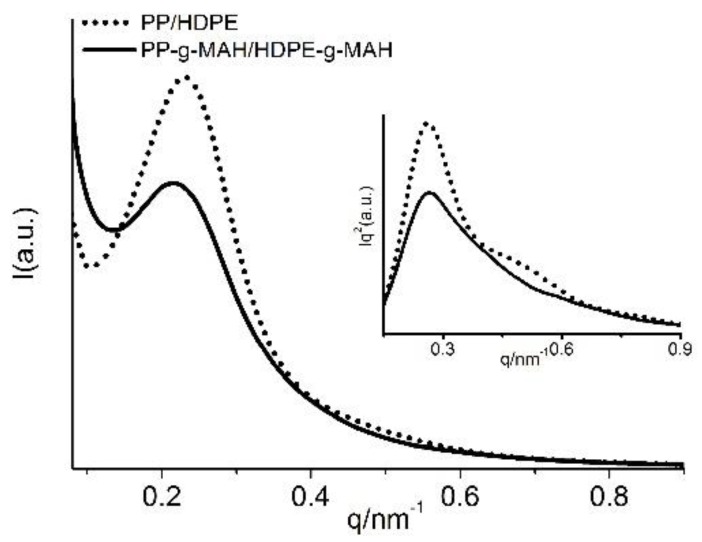
Linear and Lorentz-corrected SAXS profiles for PP/HDPE and PP-*g*-MAH/HDPE-*g*-MAH at room temperature.

**Figure 7 polymers-12-00675-f007:**
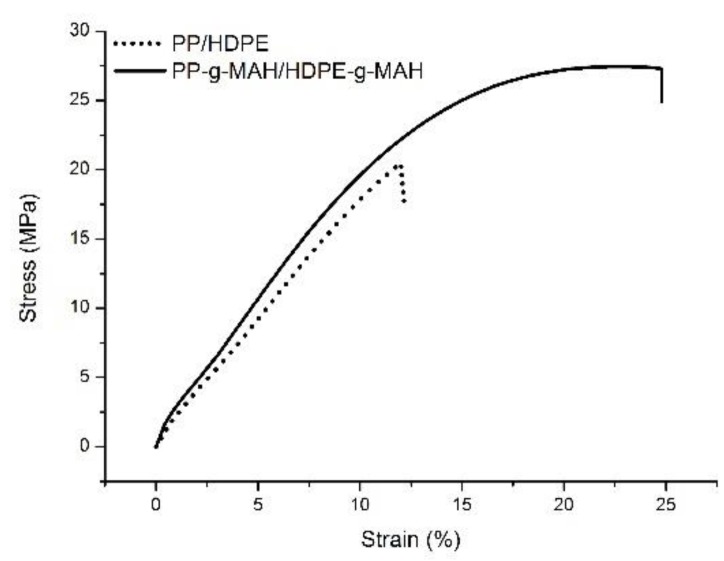
Stress and strain curves of PP/HDPE and PP-*g*-MAH/HDPE-*g*-MAH.

**Table 1 polymers-12-00675-t001:** Characteristics of the materials used.

Sample	Melt Index (g/10 min)	Grafting Degree (%)	Sample	Melt Index (g/10 min)	Density (g/cm^3^)
PP-*g*-MAH	64.3	0.75	PP	1.8	0.90
LLDPE-*g*-MAH	1.9	0.65	LLDPE	2.9	0.92
HDPE-*g*-MAH	4.4	0.65	HDPE	10.1	0.95
PP-*g*-MAH/HDPE-*g*-MAH	13.8	0.70	PP/HDPE	12.3	-----

Melting index test conditions: 230 °C, 2.16 kg.

**Table 2 polymers-12-00675-t002:** Crystal size (*L*/nm) for different polyolefin samples together with FWHW/^o^.

Sample	*FWHM* (110)	*L* (110)	*FWHM* (040)	*L* (040)	*FWHM* (130)	*L* (130)	FWHM (131)	*L* (131)
PP PP-*g*-MAH	0.64 0.64	12.36 12.36	0.56 0.57	14.29 13.95	0.63 0.64	12.91 12.37	1.28 1.29	6.25 6.19
**Sample**	***FWHM*** **(110)**		***L*** **(110)**		***FWHM*** **(200)**		***L*** **(200)**	
LLDPE LLDPE-*g*-MAH HDPE HDPE-*g*-MAH	0.60 0.61 0.48 0.51		13.35 13.21 16.61 15.72		0.71 0.72 0.52 0.55		11.29 11.07 15.36 14.48	

**Table 3 polymers-12-00675-t003:** Long period obtained from Lorentz-corrected SAXS profiles

Sample	Long Period (nm)	
	Neat Polymer	MAH Grafted
PP	15.1	12.7
HDPE	25.8	24.4
LLDPE	17.7	16.8

**Table 4 polymers-12-00675-t004:** Mechanical properties of polyolefins and grafts.

Sample	Tensile Strength (MPa)	Elongation at Break (%)	Elastic Modulus (MPa)	Yield Strength (MPa)
HDPE	27.6	2839.0	178.0	25.1
HDPE-*g*-MAH	25.3	1865.6	188.2	25.3
LLDPE	23.3	985.3	251.1	9.5
LLDPE-*g*-MAH	22.96	981.8	245.3	9.3
PP	25.6	1061.0	151.0	19.4
PP-*g*-MAH	16.6	20.2	173.3	----

**Table 5 polymers-12-00675-t005:** Mechanical properties of PP/HDPE and grafts.

Sample	Tensile Strength (MPa)	Elongation at Break (%)	Elastic Modulus (MPa)
PP/HDPE	20.5	12.0	184.0
PP-*g*-MAH/HDPE-*g*-MAH	27.5	24.8	198.0

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
