# Peer review of "Crystallization, Structures, and Properties of Different Polyolefins with Similar Grafting Degree of Maleic Anhydride"

_polymers, 2020, doi:10.3390/polym12030675_

Round 1

Reviewer 1 Report

The manuscript "Crystallization, Structures and Properties of Different Polyolefins with Similar Grafting Degree of Maleic Anhydride" by Y. Wang et al., presents a study on different maleic anhydride grafted polyolefins done by DSC, WAXD, SAXs and tensile testing. The results about the crystallization kinetics, thermal behavior, crystal structure, lamellar packing and crystal size of grafted and non-grafted polymers are reported in parallel with observations about the mechanical properties of the same materials. Although some older investigations were done on some similar systems before, the current study presents a systematic investigation of different polyolefins with similar grafting degree and relates the microscopic observations to macroscopic properties. The experimental investigation is done in a clear way, the results are well interpreted. I would like to make some suggestions regarding the presentation style: the English presentation and the text editing must be revised, the references seem incorrect in some cases (J. Appl. Polym. Sci. instead of Appl. Poly. Sci., J. Mater. Sci., instead of Mater. Sci., etc.) and the quality of Figure 5b must be improved (cooling rate and the inset).

Author Response

Reviewer #1: The manuscript "Crystallization, Structures and Properties of Different Polyolefins with Similar Grafting Degree of Maleic Anhydride" by Y. Wang et al., presents a study on different maleic anhydride grafted polyolefins done by DSC, WAXD, SAXs and tensile testing. The results about the crystallization kinetics, thermal behavior, crystal structure, lamellar packing and crystal size of grafted and non-grafted polymers are reported in parallel with observations about the mechanical properties of the same materials. Although some older investigations were done on some similar systems before, the current study presents a systematic investigation of different polyolefins with similar grafting degree and relates the microscopic observations to macroscopic properties. The experimental investigation is done in a clear way, the results are well interpreted.

I would like to make some suggestions regarding the presentation style:

Point 1: The English presentation and the text editing must be revised;the references seem incorrect in some cases (J. Appl. Polym. Sci. instead of Appl. Poly. Sci., J. Mater. Sci., instead of Mater. Sci., etc.)

Response 1: Thanks a lot for the comment and for pointing out the errors with listed references. We have carefully polished the language and corrected the mistakes with the cited references.

Point 2:  The quality of Figure 5b must be improved (cooling rate and the inset).

Response 1: We have significantly improved the quality of Fig. 5, including adding the cooling rate in figure 5 (b), as shown below for figure 5(b) for example.

Reviewer 2 Report

The paper entitled “Crystallization, Structures and Properties of Different Polyolefins with Similar Grafting Degree of Maleic Anhydride” is an interesting manuscript I suggest the publication in Polymers.

In  this  research activity, the authors studied the effects of MAH grafting on LLDPE, HDPE, PP and PP/HDPE blend. The authors observed that the grafting makes less ordered lamellar structure, a shorter long period for lamellar stacks and a smaller crystal determined by X-ray diffraction than the corresponding neat polymers. MAH grafting makes PP lamellar packing less ordered the most and almost no effect to LLDPE. The grafting leads to a smaller crystal size, less ordered lamellar structure and a shorter long period for HDPE, LLDPE and PP. The grafting makes PP lamellar packing less ordered the most and almost no effect to LLDPE. The grafting does not have that much impact on the crystallization ability of the HDPE, LLDPE and HDPE/PP blend, but appreciably reduces the crystalline ability of PP-g-MAH, due to a dramatically drop in its molecular weight during the grafting process. The authors observed that the grafting makes PP a very brittle material with a lowered average melting point than the corresponding neat PP, but the grafting has almost no effect on elongation at break for LLDPE and some effect on HDPE (decreased by one third).

In  terms of thermal degradation the PP thermal behaviour is due to MAH grafting, this trend can be avoided in the presence of PE component, i.e. making the grafting of PP and PE at the same time with HDPE/PP blend. The grafted HDPE/PP blend shows a significantly improved compatibility, which leads to overall appreciably better mechanical properties than the neat HDPE/PP blend.

The paper is very interesting the authors used different techniques to characterize the different formulations. The paper can be accepted after minor revision.

Specific comments   

Introduction section: The authors are invited to stress better the novelty of this research.    

Author Response

Reviewer #2: The paper entitled “Crystallization, Structures and Properties of Different Polyolefins with Similar Grafting Degree of Maleic Anhydride” is an interesting manuscript I suggest the publication in Polymers. 

In this research activity, the authors studied the effects of MAH grafting on LLDPE, HDPE, PP and PP/HDPE blend. The authors observed that the grafting makes less ordered lamellar structure, a shorter long period for lamellar stacks and a smaller crystal determined by X-ray diffraction than the corresponding neat polymers. MAH grafting makes PP lamellar packing less ordered the most and almost no effect to LLDPE. The grafting leads to a smaller crystal size, less ordered lamellar structure and a shorter long period for HDPE, LLDPE and PP. The grafting makes PP lamellar packing less ordered the most and almost no effect to LLDPE. The grafting does not have that much impact on the crystallization ability of the HDPE, LLDPE and HDPE/PP blend, but appreciably reduces the crystalline ability of PP-g-MAH, due to a dramatically drop in its molecular weight during the grafting process. The authors observed that the grafting makes PP a very brittle material with a lowered average melting point than the corresponding neat PP, but the grafting has almost no effect on elongation at break for LLDPE and some effect on HDPE (decreased by one third).

In terms of thermal degradation, the PP thermal behavior is due to MAH grafting, this trend can be avoided in the presence of PE component, i.e. making the grafting of PP and PE at the same time with HDPE/PP blend. The grafted HDPE/PP blend shows a significantly improved compatibility, which leads to overall appreciably better mechanical properties than the neat HDPE/PP blend.

The paper is very interesting the authors used different techniques to characterize the different formulations. The paper can be accepted after minor revision.

Point 1: Introduction section: The authors are invited to stress better the novelty of this research.

Response 1: We appreciate the reviewer’s comments. In term of the novelty/Introduction of this work, we have made significant changes for the Introduction of the revised manuscript, including

  1. Besides the focus on the different effects of similar MAH grafting degree on crystallization, structures and properties of different polyolefins, which has not been investigated, a very different conclusion is made about the double melting points of PP-g-MAH, based on our studies with multiple techniques, including degradation characterization. In literature, this double peak feature is misinterpreted as the perfection of the crystal structure during the melting process.
  2. Added more references to the Introduction, such as, references 16 for Maleic anhydride/ styrene-grafted PPR (Polypropylene Random Copolymer).
  3. Polished the language very carefully, including corrected some errors
